# Two-dimensional amine and hydroxy functionalized fused aromatic covalent organic framework

Javeed Mahmood [1], Ishfaq Ahmad[1], Minbok Jung[2], Jeong-Min Seo[1], Soo-Young Yu[1], Hyuk-Jun Noh[1], Young Hyun Kim[1], Hyung-Joon Shin [2✉] & Jong-Beom Baek [1✉]

Ordered two-dimensional covalent organic frameworks (COFs) have generally been synthesized using reversible reactions. It has been difficult to synthesize a similar degree of ordered COFs using irreversible reactions. Developing COFs with a fused aromatic ring system via an irreversible reaction is highly desirable but has remained a significant challenge. Here we demonstrate a COF that can be synthesized from organic building blocks via irreversible condensation (aromatization). The as-synthesized robust fused aromatic COF (F-COF) exhibits high crystallinity. Its lattice structure is characterized by scanning tunneling microscopy and X-ray diffraction pattern. Because of its fused aromatic ring system, the F-COF structure possesses high physiochemical stability, due to the absence of hydrolysable weak covalent bonds.

[1] School of Energy and Chemical Engineering, Center for Dimension-Controllable Organic Frameworks, Ulsan National Institute of Science and Technology (UNIST), 50 UNIST, Ulsan 44919, Republic of Korea. [2] School of Materials Science and Engineering & Low-Dimensional Carbon Materials Center, Ulsan National Institute of Science and Technology (UNIST), 50 UNIST, Ulsan 44919, Republic of Korea. ✉email: shinhj@unist.ac.kr; jbbaek@unist.ac.kr

Covalent organic frameworks (COFs) are a class of crystalline macromolecular structures whose periodically linked building units extend into a framework with uniform topology and pores[1–4]. Unlike other more widely studied crystalline materials (e.g., graphene[5] and boron nitride[6]), the COFs have a unique aspect in that their skeletons and pores can be precisely engineered, like those of metal organic frameworks (MOFs)[1,7–9]. Two-dimensional (2D) COFs can be divergently expanded into a π-conjugated structure by periodically integrating the building blocks[9–12]. Because of their ordered nature and 2D topology, COFs are seen as a dynamic and robust platform for the design of advanced functional materials, including a wide range of semiconductors[11], and proton conductors[13,14] for gas adsorption[15–17], catalysis[18–21] and energy conversion and storage applications[22–25]. The ability to generate crystalline organic materials with precise control of the framework at a molecular-level has been one of the most important recent developments in chemistry and materials science[1,8,9].

In order to achieve high crystallinity in COFs, current methods typically rely on unstable bond formation using a reversible reaction to link monomer units by thermodynamic equilibria[9,26]. However, this beneficial intrinsic reversibility limits the COFs practical applications, because it leads to physiochemical instability[4,27]. COFs with boroxine or boronate linkages are prone to amorphization or disintegration in water or protic solvents[4,27,28].

The prevailing class of COFs with relatively stable linkages, such as imine (-C=-) based COFs, exhibit enhanced hydrothermal stability[29]. Because of their reversible nature, however, under vigorous acidic conditions the chemical stability of most imine based COFs against hydrolysis is far from satisfactory[30]. To address the stability issue, and enable the fabrication of physiochemically robust COFs, a number of methodologies have been explored[31–35]. However, further exploration of new reticular chemistry is needed to synthesize stable COFs as reliable functional materials, and crucial to widen their practical applications.

The intrinsically important feature of COFs is their ability to form covalently linked stable aromatic ring systems, which are the basis of their exceptional electronic and magnetic properties[36]. In COFs, π–π interactions form porous layered frameworks[3,27,37]. However, typical COFs lack π–π interlayer interactions, and chemical stability, which limits their applications. Imine and boroxine based COFs, for example, are inferior when it comes to promoting π electron delocalization between the connecting units[38]. Despite a few examples of π-conjugated COFs, obtained using surface or interface-assisted synthesis[10,39,40], the design and synthesis of fused aromatic ring-based π-conjugated COFs, which

produces high crystallinity and structural stability in corrosive environments, remains a crucial challenge[41].

To tackle this issue, a physiochemically robust yet fully conjugated fused aromatic COFs is highly desired, particularly to address long-standing challenges in semiconductor technology[11,42].

Here, we report a fully conjugated fused aromatic 2D COF with aromatic amine (−NH₂) and hydroxyl (−OH) functionalities in the pores. The fused aromatic COF allows inherently periodic ordering with extended π-electron delocalization and thus physiochemical stability. The process relies on the powerful polycondensation (aromatization) of pentaaminophenol (PAP) and hexaketocyclohexane (HKH) in trifluoromethanesulfonic acid (TFMSA) to yield fused aromatic phenazine-linked 2D COF (denoted F-COF, where F stands for fused, indicating the formation of fused aromatic pyrazine rings after the reaction between the *ortho*-diketone and *ortho*-diamine moieties).

## Results and discussion

**Synthesis and characterization of the F-COF.** Figure 1 demonstrates the physiochemically stable structure of the entirely aromatic π-conjugated COF. PAP (Supplementary Methods) was chosen as a pseudo-$C_2$-symmetric monomer, and HKH was chosen as a $C_3$-symmetric monomer. Due to the formation of fused pyrazine rings in the network-forming reaction, crystalline F-COF was produced in quantitative yield even in solution, without surface and/or interfacial assistance (Fig. 1). The mixing of the monomers was carried out in TFMSA at −40 °C (melting temperature) to slow down the reaction kinetics between PAP and HKH to increase crystallinity. Thus far, a number of pyrazine-based crystalline COF structures have been reported by solution and solvothermal processes[11,42–46]. The F-COF was systematically characterized using various analytical techniques. Elemental analysis (EA) confirmed the formation of structure and exhibited experimental values that were very close to the theoretical values (Supplementary Table 1). Thermogravimetric analysis (TGA) indicated that the F-COF has good thermal stability in both nitrogen and air environments (Supplementary Fig. 1).

The qualitative bonding nature in F-COF was analyzed using X-ray photoelectron spectroscopy (XPS). The XPS survey spectra revealed the presence of only three peaks, related to carbon (C 1s), nitrogen (N 1s), and oxygen (O 1s) (Supplementary Fig. 2a). The deconvoluted C 1s spectrum gave three peaks at 284.08, 285.31, and 286.74 eV, which are assignable to sp2 C–C, sp2 C–N, and C–OH/C–NH₂, matching the structure well (Supplementary Fig. 2b). The high-resolution N 1s has two peaks

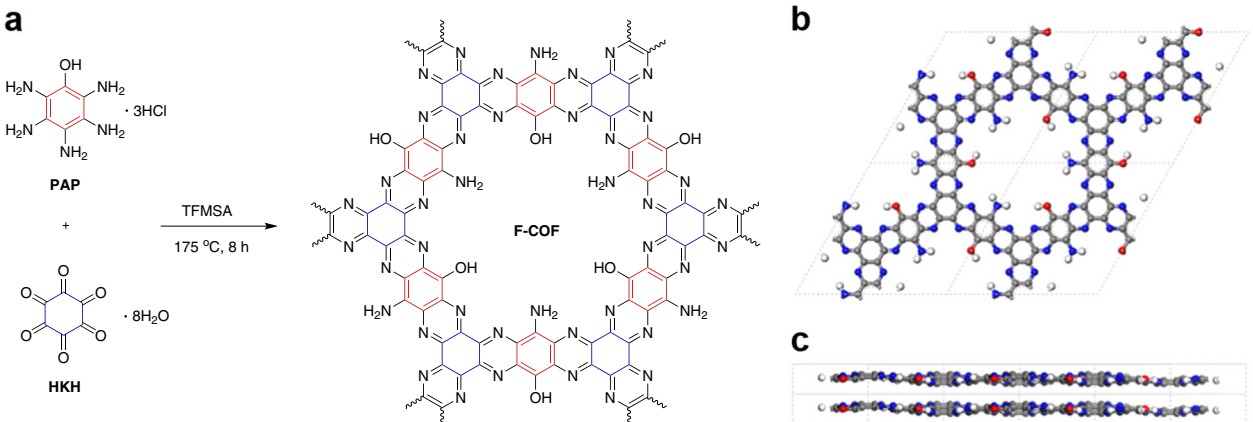

**Fig. 1 Schematic illustration of the synthesis and structure of F-COF. a** The formation of F-COF from the reaction between pentaaminophenol (PAP) trihydrochloride and hexaketocyclohexane (HKH) in freshly distilled trifluoromethanesulfonic acid (TFMSA). Extended energy minimized eclipsed structures of F-COF: **b** top view; **c** side view. Color codes: C, gray; H, white; N, blue; O, red.

at 398.55 and 399.97 eV corresponding to pyrazine-like nitrogen ($sp2$ N–C) and C–NH$_2$ (Supplementary Fig. 2c). The deconvoluted oxygen (O 1$s$) in the structure is assigned to residual C=O (531.49 eV) at the edges and the phenolic C–OH (532.96 eV) in the structure (Supplementary Fig. 2d).

The formation of the F-COF was also detected by Fourier-transform infrared (FT-IR) spectroscopy, which revealed the aromatic C=N, C=C stretching vibration at 1627 cm$^{-1}$. Furthermore, the band at 1465 cm$^{-1}$ is related to aromatic ring stretching vibration. The bands at 1387 and 1254 cm$^{-1}$ can be attributed to C−N and C−O stretching vibrations. The band from 3129 to 3427 cm$^{-1}$ is associated with the stretching vibration of the O–H and N–H bonds in the structure (Supplementary Fig. 3). The solid-state carbon 13 cross-polarization-magic angle spinning nuclear magnetic resonance ($^{13}$C CP-MAS NMR) spectroscopy was exploited to investigate the chemical structure of the F-COF. Two intense peaks centered at 138.50 and 176.40 ppm are, respectively, related to the carbon atoms attached to nitrogen and edge ketonic groups (Supplementary Fig. 4). The bulk microstructure of the F-COF was studied with field-emission scanning microscopy (FE-SEM). SEM image showed the 2D layered morphology with a grain size of few tens of micrometers (Supplementary Fig. 5). High-resolution transmission electron microscopy (HR-TEM) image also displayed sheet like texture (Supplementary Fig. 6). However, due to multi-layer stacking and beam damage, resolving structure was not possible[11,43–45].

The crystalline nature of F-COF was resolved by powder X-ray diffraction (PXRD) analyses in combination with theoretical XRD simulations. The peak at 6.42° (2$\theta$) can be assigned to the (100) plane of a crystalline hexagonal arrangement. The relatively strong peak at 27.05° is assignable to the (001) plane, which is related to the interlayer π–π stacking. The relatively broad PXRD peaks of the F-COF are associated with the less-ordered edges and extremely large molecular size of the F-COF along with poor stacking due to irreversibility of the reaction[3,11,47,48] (Supplementary Note 1). The crystalline structure of F-COF was determined using XRD simulation and Pawley refinement in combination with experimental PXRD patterns (Fig. 2a).

Structure based on a hexagonal lattice in the space group $P3$ was selected for F-COF.

The unit cell parameters and simulated PXRD patterns were acquired ($a = b = 16.6$ Å, $c = 3.30$ Å) using the geometrical energy minimization of the structure by a universal force field. The PXRD pattern of the F-COF coincided well with the AA-stacking model (Fig. 2b), while the AB-stacking model deviated from the experimentally obtained profile (Fig. 2c). Figure 2a displays a comparison of the Pawley-refined XRD curve with the experimental one, which has small differences ($R_{wp} = 3.48\%$, $R_p = 5.44\%$). These results demonstrate the hexagonal ordering of the extended 2D structure along the $x$ and $y$ directions with 3.30 Å layer to layer distance in the $z$-axis.

**Scanning tunneling microscopy imaging of the F-COF structure**. A scanning tunneling microscopy (STM) study was performed to visualize the atomic structure of the F-COF. A single sheet of F-COF was prepared on a Cu(111) substrate under ultrahigh vacuum (UHV) by thermal evaporation at 600 K. The STM measurements were performed in UHV at a low temperature (77 K). Figure 3a is a high-resolution STM image of the F-COF on the Cu(111) substrate, showing the precise holey structure of the hexagonal array (Supplementary Note 2). The hole-to-hole distance, determined from the height profiles, was approximately 15.10 ± 0.15 Å, which matched well with the theoretical hole-to-hole distance.

The electronic structure of the F-COF was examined with scanning tunneling spectroscopy (STS) using lock-in detection mode. Two broad peaks appeared at −1.41 and 0.59 eV in the valence band and conduction band zones, respectively (Fig. 3c). The direct bandgap of the F-COF was empirically determined by Tauc plot from ultraviolet–visible (UV) spectroscopy (Supplementary Fig. 7). The value was found to be 2.00 eV, agreeing well with the STS result.

As the solution synthesized materials by polymerization have a wide range of molecular weight distribution from small to very large flakes sizes. It is almost impossible to synthesize macromolecular materials with the same molecular size and uniform

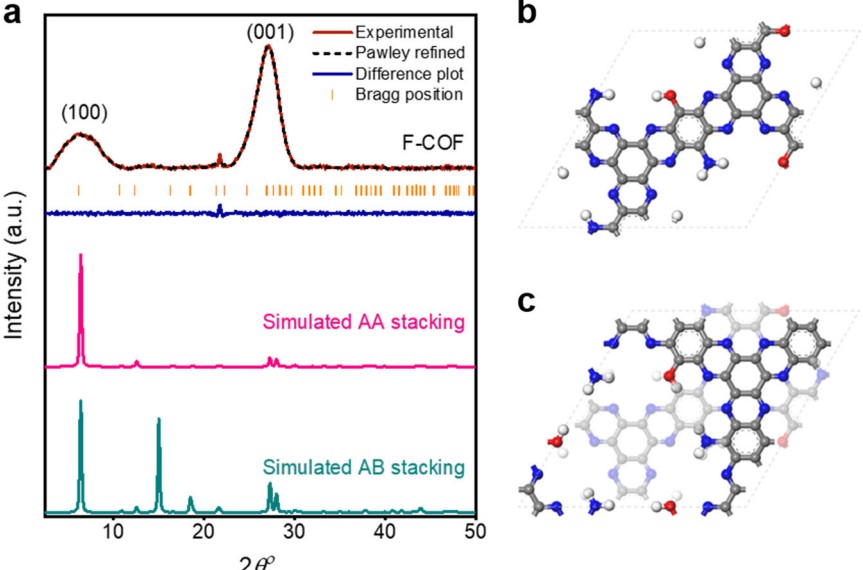

**Fig. 2 Powder X-ray diffraction study of F-COF. a** Experimental (red line), Pawley-refined pattern (black dash line), Bragg position (orange line), the difference plot (blue line), simulated AA-stacking model (pink line) and AB-stacking model (green line). **b** Crystallographic unit cells of F-COF with eclipsed AA-stacking model. **c** Staggered AB-stacking model. Color codes: C, gray; H, white; N, blue; O, red.

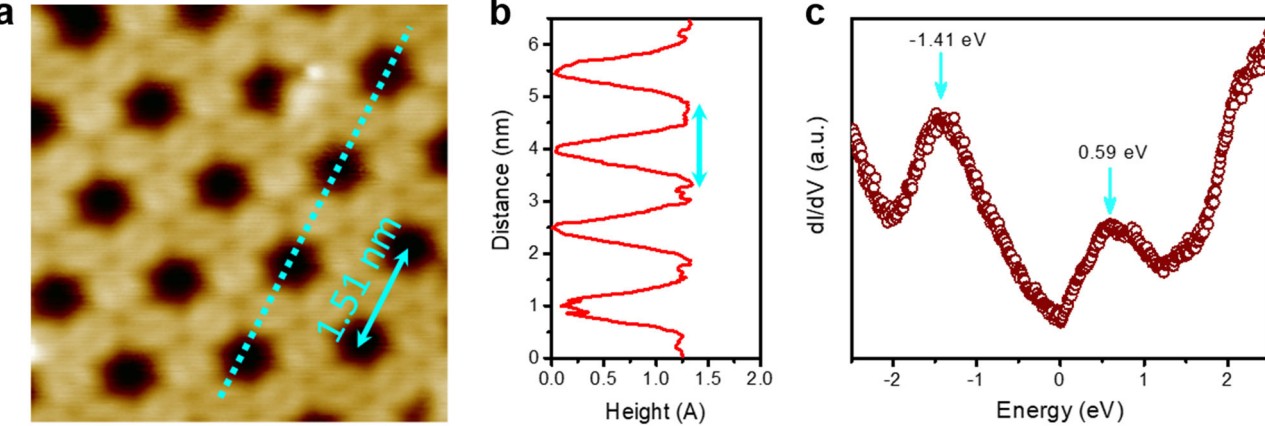

**Fig. 3 Scanning tunneling microscopy analysis and theoretical calculations. a** An atomic-resolution scanning tunneling microscopy (STM) topographic structure of F-COF on Cu(111) substrate. The STM image (5.4 × 5.4 nm$^2$) was acquired at a sample bias of –0.2 V and a tunneling current of 20 pA. **b** Topographic height profile along the cyan blue dotted line, indicating a hole-to-hole distance of 1.51 nm. **c** Differential conductance (d*I*/d*V*) spectrum of F-COF.

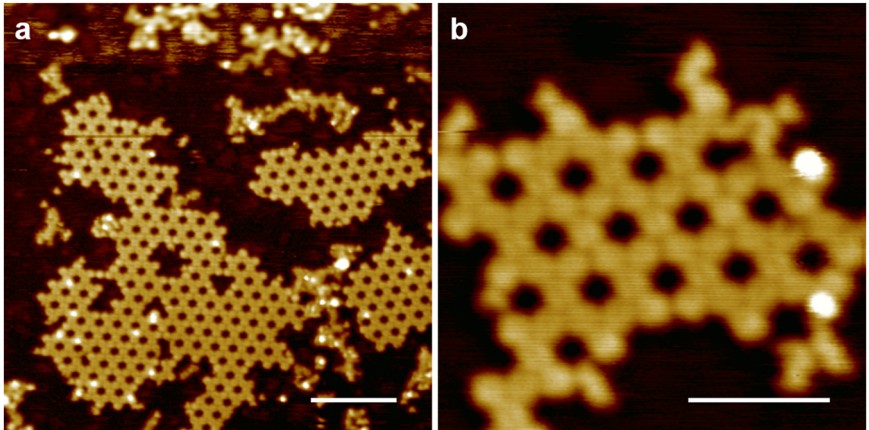

**Fig. 4 STM images showing different sizes of F-COF flakes. a** STM image (36 × 36 nm$^2$) acquired at a sample bias of –1.0 V and a tunneling current of 50 pA. **b** Image (9 × 9 nm$^2$) acquired at a sample bias of –1.5 V and a tunneling current of 50 pA. Scale bar: **a** 7 nm; **b** 3 nm.

molecular weight distribution. Thus, under UHV condition, it is possible to sublimate relatively smaller size flakes onto the cleaned Cu(111) substrate. From the STM we were able to observe different sizes of flake (Fig. 4) on Cu(111) substrate. During the STM characterization, we managed to see a small piece consisting of just two holes of the F-COF structure (Supplementary Fig. 8).

The stability of the F-COF was investigated by soaking the sample for 24 h in concentrated hydrochloric acid (HCl) and 6 M sodium hydroxide (NaOH). The PXRD results clearly indicate stability of the structure, showing no changes in the peak position before and after acid and base treatment (Supplementary Fig. 9).

In summary, we designed and synthesized one of the few pyrazine-based stable COFs[11,42] reported to date via a wet-organic reaction. This physiochemically stable F-COF consists of all aromatic rings with a π-conjugated network structure and tunable functional groups. In contrast, the previously reported pyrazine-based COFs have no available functional groups[11,42]. The structure of the F-COF is visualized at atomic level by STM study, which reveals there are the same number of two different kinds of functional groups (−NH$_2$ and −OH) in each hole. This suggests there is further modification potential (Supplementary Fig. 10) for numerous specific applications. The design and synthesis of F-COF may help in the synthesis of other tunable

COFs with fully π-conjugated stable aromatic systems. We firmly anticipate these kinds of functionalized COFs can expedite further innovations and applications in the COF field, from wet-chemistry to device applications, such as organic electronics, energy conversion, and storage.

## Methods

**Synthesis of functionalized two-dimensional framework (F-COF).** PAP (2 g, 7.17 mmol) was charged in a three-necked round bottom flask under argon atmosphere and placed in a cold bath at −40 °C, and freshly distilled tri-fluoromethanesulfonic acid (TFMSA, 40 ml) was added. Then, HKH (1.49 g, 4.78 mmol) was slowly added while stirring at −40 °C for 4 h. The reaction flask was slowly allowed to warm up to room temperature. The ice bath was replaced with oil bath and heated to 175 °C for 8 h. Then, the flask was cooled to room temperature and poured into water. The solid product that precipitated was collected by suction filtration using a polytetrafluoroethylene (PTFE, 0.5 μm) membrane. The resultant dark solid was further Soxhlet extracted with methanol and water, respectively, for 3 days each and freeze-dried at −120 °C under reduced pressure (0.05 mm Hg) for 3 days.

**STM experiments.** The STM experiments were performed in low-temperature HV at 77 K (SPECS JT-STM). The cleaned single crystal Cu(111) surface was prepared with a few cycles of Ar$^+$ sputtering and annealing. After obtaining a cleaned Cu(111) substrate, an F-COF monolayer was deposited on the precleaned Cu(111) substrate by in situ thermal evaporation under UHV condition. The temperature of the Cu(111) was kept at room temperature and the F-COF evaporation tempera-ture was about 600 K.

**Characterization**. All the chemicals, reagents, and solvents were purchased from Aldrich Chemical Inc., unless otherwise stated. Solvents were degassed with nitrogen purging prior to use. All reactions were accomplished under nitrogen atmosphere using oven dried glassware.

EA was performed with a Thermo Scientific Flash 2000 Analyzer. Proton ($^1$H) and carbon thirteen ($^{13}$C) nuclear magnetic resonance (NMR) spectra were recorded on an AVANCE III HD 400 MHz FT-NMR (Bruker) spectrometer for the monomer characterization. Solid-state NMR spectrum of F-COF was measured using powder sample on 600 MHz VARIAN FT-NMR (Agilent) at a spinning rate of 20 kHz. NMR spectra can be found in Supplementary Figs. 11–17. Melting points were measured on a KSP1N automatic melting point meter (A. Krüss Optronic GmbH, Germany). High-resolution mass spectra (HRMS) were measured using JEOL/JMS-700. XPS was performed on an X-ray Photoelectron Spectrometer Thermo Fisher K-alpha (UK). X-ray diffraction (XRD) studies were taken on a High-Power X-Ray Diffractometer D/MAZX 2500 V/PC (Cu-Kα radiation, 35 kV, 20 mA, $\lambda = 1.5418$ Å) Rigaku, Japan at 40 kV and 200 mA at room temperature. Scanning electron microscope (SEM) images were obtained with Pt- coated samples on carbon tape by a Field Emission Scanning Electron Microscope Nanonova 230 (FEI Inc., USA). HR-TEM images were taken by a JEM-2100F microscope (JEOL inc., Japan) under an operating voltage of 200 keV. The samples were prepared by drop casting of dispersed ethanol on holey carbon TEM grid and dried in oven at 50 °C under vacuum. The TGA was carried using a Thermogravimetric Analyzer Q200 (TA Instrument, USA) at a heating rate of $10^o$ min$^{-1}$ in nitrogen and dry air atmosphere. Fourier-transform infrared (FT-IR) spectra were conducted on a Spectrum 100 (Perkin-Elmer, USA) with KBr pellet. XRD simulation and Pawley refinement were carried out using Materials studio modeling V. 7.0 (Accelrys, 2013). Pawley refinement was carried out to optimize the lattice parameter iteratively. The pseudo-Voigt function was exploited for whole profile fitting and the Berrar–Baldinozzi function was employed for to correct asymmetry during the refinement processes, until the $R_p$ and $R_{wp}$ values converged.

## Data availability

All principal data with detailed experimental procedure and characterization of this work are included in this article and its Supplementary Information or are available from the corresponding author upon reasonable request.

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

## Acknowledgements

This work was supported by the Creative Research Initiative (CRI, 2014R1A3A2069102), BK21 Plus (10Z20130011057), Science Research Center (SRC, 2016R1A5A1009405), Basic Science Research (2018R1A2B6006423) and Young Researcher (2019R1C1C1006650) Programs through the National Research Foundation (NRF) of Korea, funded by the Ministry of Science, ICT, and Future Planning.

## Author contributions

J.-B.B. and J.M. conceived and designed the project. J.M., I.A., J.-M.S., S.-Y.Y., H.-J.N., and Y.H.K., carried out the synthesis and characterizations of monomers and organic framework. M.J. and H.-J.S. studied the STM characterization. J.M., H.-J.S. and J.-B.B. wrote the manuscript and all authors discussed the results and commented on the paper.

## Competing interests

The authors declare no competing interests.
