## [Peer Review File · Communications Chemistry]

Reviewers' comments:

Reviewer #1 (Remarks to the Author):

This paper describes an interesting structural analog of an aza-fused covalent organic framework with good crystallinity derived from the condensation pentaaminophenol and hexaketocyclohexane. The manuscript represents an important advance in aza-fused framework synthesis, but has several shortcomings that should be carefully addressed by the authors:

1. The statement put forward by the authors "Thus far, only two pyrazine based crystalline COF structures have been reported: one was prepared by solution process¹¹ and the other employed solvothermal synthesis.⁴²" is incorrect as it overlooks several significant developments in this class of materials listed below. The authors should be careful to conduct a thorough literature review to ensure that their claims are correct and carefully account for and acknowledge the recent developments in the field.

Kou, Y.; Xu, Y.; Guo, Z.; Jiang, D. Supercapacitive Energy Storage and Electric Power Supply Using an Aza-Fused π -Conjugated Microporous Framework. *Angew. Chem., Int. Ed.* 2011, 50, 8753– 8757, DOI: 10.1002/anie.201103493

Meng et al. "Proton Conduction in 2D Aza-Fused Covalent Organic Frameworks" *Chem. Mater.* 2019, 31, 819-825., DOI: 10.1021/acs.chemmater.8b03897

Meng et al. "Two-Dimensional Chemiresistive Covalent Organic Framework with High Intrinsic Conductivity" *J. Am. Chem. Soc.* 2019, 141, 11929-11937. DOI: 10.1021/jacs.9b03441

Wang et al. "Unveiling Electronic Properties in Metal-Phthalocyanine-Based Pyrazine-Linked Conjugated Two-Dimensional Covalent Organic Frameworks" *J. Am. Chem. Soc.* 2019, 141, 16810-16816. DOI: 10.1021/jacs.9b07644.

2. In light of the previous comment, the title of the work is too generic, given that there are several other reports of aza-fused frameworks available. The authors should make their title more specific and descriptive to reflect their specific contribution to this field.

3. The synthetic scheme presented in Figure 1 should list all of the reagents, additives, and solvents and key reaction conditions (such as temperature, pressure, reaction time, etc) for the optimized chemical transformation shown to quickly highlight these aspects for the reader. Furthermore, a detailed procedure for the COF synthesis should be included into the SI. As it stands, the article currently does not give an experimental procedure for the synthesis of the COF material that would be sufficient for reproducing the work.

4. The pXRD spectrum of the framework material shown in Figure 2 exhibits very broad peaks, where relative peak heights between experimental and simulated spectra are not in full agreement. The authors should take care to rationalize deviations in relative peak heights obtained experimentally from the simulated spectrum. The authors should also comment to acknowledge the reasons behind the broadness of peaks in the spectrum (poor crystallinity, small crystal size, etc.) in more detail. In particular, it is unclear what the authors mean by "extremely large molecular size of the F-COF" as rationale for the broadness in peaks in the context of the Scherrer equation relating crystallite size to peak broadening.

5. Did the authors undertake any reaction optimization for obtained the fused COF material? It is quite common to implement extensive reaction optimization to achieve and optimize COF synthesis. Any reaction optimization work should be reported, and comments should be made about important experimental steps, additives, etc that are critical to obtaining crystalline material.

6. Figure 1 seems to suggest an alternating arrangement of -OH and -NH₂ moieties within each pore of the framework. The authors should comment what evidence (if any) this structural model is based on and if other arrangements are possible. Furthermore, some rationale for the molecular design of these functional group moieties should be provided. Is -OH substitution expected to provide certain synthetic or functional advantages over hexamine substitution?

7. The authors should take care to describe which instruments (brand, make, model) of each instrument were used to obtain various characterizations. All key settings of the experimental conditions and sample preparation for analysis should be given. This information is essential for ensuring the reproducibility of the work and the utility of this work to others.

8. Characterization of the material by SEM and TEM would be helpful in understanding the bulk morphology and long range order of the framework. STM only show small fragments and do not help in ascertaining long-range order or bulk morphology of the material.

9. The text in the experimental section of SI "Important Note regarding STM" should be refined to make it more clear and improve English language usage. How are the samples prepared for STM? Is there sonication and dropcasting involved? What are the key instrument settings for obtaining the images?

10. What is the red trace in Figure S1? Please label.

Reviewer #2 (Remarks to the Author):

This work describes the formation of pyrazine consists of a reversible amine/ketone addition and an irreversible ring closure. The value of the work is that the product is much more conjugated than normal Schiff-base COF, so this represents a significant advancement.

One point that needs attention is that the crystallinity is more like a polymer from powder XRD. Can the authors comment on this? compared to the usual COF made the XRD peaks of this is broad, should it be closer to a polymer when crystallized in bulk form.

However on-surface synthesis shows that 2-D COF can be formed locally. The STM result is reasonably good among all published 2D polymer grown in STM, in terms of uniformity and size. Such dehydration reaction is usually not suitable for on-surface growth, so this result is excellent and the work can be published. At least, it provides a design guiding principle for future researchers.

Reviewer #3 (Remarks to the Author):

In this manuscript, the authors synthesize a covalent organic framework featuring fused aromatic rings via irreversible condensation. This pyrazine-based COF has been characterized with different techniques (PXRD, TGA, XPS, FT-IR, STM etc.) and exhibits a high physicochemical stability owing to its fused aromatic ring system.

After reading through the whole manuscript carefully, I think it is not suitable to publish in Communications Chemistry as it does not meet the novelty criteria and the studies are not sufficient.

1. Compared with other pyrazine-based COFs (J. Am. Chem. Soc. 2019, 141, 16810., ref. 42), the

quality of PXRD pattern is poor, and the values of R_{wp} and R_p are not reasonable.

2. Compared with the paper that the authors published in Nat. Commun. previously, this manuscript does not have too much advantage and novelty.

3. COF is crystalline polymer and not easy to sublime. For the STM experiment, the authors have claimed that "They rigorously evaporate and sublime.". Why? Are they oligomers? Or they are the mixture of COFs and oligomers?

4. The Solid state NMR of the COF should be provided to analyze the structure of F-COF.

5. There are some mistakes in the manuscript and SI, such as Supplementary Fig. 1, weight loss is wrong.

Reviewers' comments:

Reviewer #1 (Remarks to the Author):

This paper describes an interesting structural analog of an aza-fused covalent organic framework with good crystallinity derived from the condensation pentaaminophenol and hexaketocyclohexane. The manuscript represents an important advance in aza-fused framework synthesis, but has several shortcomings that should be carefully addressed by the authors:

Comment 1.1. The statement put forward by the authors “Thus far, only two pyrazine based crystalline COF structures have been reported: one was prepared by solution process¹¹ and the other employed solvothermal synthesis.⁴²” is incorrect as it overlooks several significant developments in this class of materials listed below. The authors should be careful to conduct a thorough literature review to ensure that their claims are correct and carefully account for and acknowledge the recent developments in the field.

Kou, Y.; Xu, Y.; Guo, Z.; Jiang, D. Supercapacitive Energy Storage and Electric Power Supply Using an Aza-Fused π -Conjugated Microporous Framework. *Angew. Chem., Int. Ed.* 2011, 50, 8753– 8757, DOI: 10.1002/anie.201103493

Meng et al. “Proton Conduction in 2D Aza-Fused Covalent Organic Frameworks” *Chem. Mater.* 2019, 31, 819-825., DOI: 10.1021/acs.chemmater.8b03897

Meng et al. “Two-Dimensional Chemiresistive Covalent Organic Framework with High Intrinsic Conductivity” *J. Am. Chem. Soc.* 2019, 141, 11929-11937. DOI: 10.1021/jacs.9b03441

Wang et al. “Unveiling Electronic Properties in Metal–Phthalocyanine-Based Pyrazine-Linked Conjugated Two-Dimensional Covalent Organic Frameworks” *J. Am. Chem. Soc.* 2019, 141, 16810-16816. DOI: 10.1021/jacs.9b07644.

Response 1.1. We highly appreciate the reviewer #1 for thoughtful evaluation and valuable suggestions to improve the literature survey and quality of the manuscript. We are sorry for missing some important references related to this work. As a matter of fact, the manuscript was written before the recent pointed references were published online. We have now added the underlined references in the revised manuscript with corrections of the related text.

Comment 1.2. In light of the previous comment, the title of the work is too generic, given that there are several other reports of aza-fused frameworks available. The authors should make their title more specific and descriptive to reflect their specific contribution to this field.

Response 1.2. We agree with the reviewer and based on the reviewer’s suggestion; we have modified the title (**Two-dimensional amine and hydroxy functionalized fused aromatic covalent organic framework**) to make it more understandable for the readers.

Comment 1.3. The synthetic scheme presented in Figure 1 should list all of the reagents, additives, and solvents and key reaction conditions (such as temperature, pressure, reaction time, etc) for the optimized chemical transformation shown to quickly highlight these aspects for the reader. Furthermore, a detailed procedure for the COF synthesis should be included into the SI. As it stands, the article currently does not give an experimental procedure for the synthesis of the COF material that would be sufficient for reproducing the work.

Response 1.3. Thanks to the reviewer for the constructive suggestion. We made necessary changes in Figure 1 and added the reaction conditions. Detailed experimental procedure is given in the Methods section. The synthesis procedures related to monomer are given in the SI along with characterizations.

Comment 1.4. The pXRD spectrum of the framework material shown in Figure 2 exhibits very broad peaks, where relative peak heights between experimental and simulated spectra are not in full agreement. The authors should take care to rationalize deviations in relative peak heights obtained experimentally from the simulated spectrum. The authors should also comment to acknowledge the reasons behind the broadness of peaks in the spectrum (poor crystallinity, small crystal size, etc.) in more detail. In particular, it is unclear what the authors mean by “extremely large molecular size of the F-COF” as rationale for the broadness in peaks in the context of the Scherrer equation relating crystallite size to peak broadening.

Response 1.4. We agree with the reviewer that the XRD spectrum reveals broad peaks. The broad XRD peaks are very common in the irreversible reaction driven synthesis of organic framework. With fast growing molecular weight in solution because of large energy gain (aromatization), it is difficult to regularly stack in a long-range planarly ordered manner due to difficulty in the movement (kinetics) of the large flakes with ripples and wrinkles. We also agree with the reviewer the broad spectrum could be related to poor crystallinity and small crystal size. However, in this case, from the SEM, TEM and STM (only small size was delaminated and observed images), we can see the size of the flakes formed large enough due to huge thermodynamic energy gain (aromatization).

Straight forward application of Scherrer equation, which is limited for nano-scale crystallites. For polymer structure, it is not very simple due to polycrystallinity. Because the shape of crystallites is usually irregular, most of the applications of the Scherrer analysis assume spherical crystallite shapes. The peak broadening can result from the non-uniform lattice distortions, dislocations, mixture of crystalline phases and grain boundaries. It is also suggested that the internal pressure exerted by the surface tension on the nanomaterial will create a stress field to trigger lattice strain, in case of small crystallite size (*Journal of Theoretical and Applied Physics* **2014**, 8, 123-134).

Comment 1.5. Did the authors undertake any reaction optimization for obtained the fused COF material? It is quite common to implement extensive reaction optimization to achieve and optimize COF synthesis. Any reaction optimization work should be reported, and comments should be made about important experimental steps, additives, etc that are critical to obtaining crystalline material.

Response 1.5. Capitalizing on the experience we already had in the synthesis of irreversible reaction based organic frameworks, we were able to synthesize this material. One thing we found was the addition of second monomer at low temperature to control the very quick reaction between the diketone and diamine to form pyrazine type rings. In this case, TFMSA (melting point: $-40\text{ }^{\circ}\text{C}$) was used as a solvent and cooling bath of $-40\text{ }^{\circ}\text{C}$ was employed to increase the crystallinity by controlling the temperature. No other optimizations were conducted in this regard.

Comment 1.6. Figure 1 seems to suggest an alternating arrangement of $-\text{OH}$ and $-\text{NH}_2$ moieties within each pore of the framework. The authors should comment what evidence (if any) this structural model is based on and if other arrangements are possible. Furthermore, some rationale for the molecular designed of these functional group moieties should be provided. Is $-\text{OH}$ substitution expected to provide certain synthetic or functional advantages over hexamine substitution?

Response 1.6. Thanks to the reviewer for the insightful comment and pointing out very important issue. It was really hard to differentiate the ($-\text{NH}_2$ and $-\text{OH}$) functional groups by any means including STM, as their van der Waals radii are very similar each other. Occurrence of ($-\text{NH}_2$ and $-\text{OH}$) functional groups is important for selective functionalization of the material for further applications. Like attaching long alkyl chain for better solubility in different organic solvents and to modulate the electronic properties of the material (**Figure R1**).

Figure R1 | Possibility of the functionalization of ($-\text{NH}_2$ and $-\text{OH}$) functional groups.

Comment 1.7. The authors should take care to describe which instruments (brand, make, model) of each instrument were used to obtain various characterizations. All key settings of the experimental conditions and sample preparation for analysis should be given. This information is essential for ensuring the reproducibility of the work and the utility of this work to others.

Response 1.7. The elaborated details of each instrument along with brand and model are added in the characterization section of the revised manuscript.

Comment 1.8. Characterization of the material by SEM and TEM would be helpful in understanding the bulk morphology and long-range order of the framework. STM only show small fragments and do not help in ascertaining long-range order or bulk morphology of the material.

Response 1.8. Thanks to the reviewer for the constructive suggestion. We have given the SEM and TEM analysis for the reviewer's consideration. The SEM analysis of the F-COF displayed the 2D structural arrangements (**Figure R2**), showing the stacking of the layers. As the reviewer may understand, it is very difficult to check the crystallinity of porous COFs by TEM, in which the electron beam quickly deforms their structures (beam damages) and the images look like less crystalline (**Figure R3**). Moreover, due to poor stacking because of the irreversibility of the reaction, it was not possible to resolve its structure using TEM (*Nat. Commun.* 2015, 6, 6486; *Chem. Mater.* 2019, 31, 819–825; *J. Am. Chem. Soc.* 2019, 141, 11929–11937; *J. Am. Chem. Soc.* 2019, 141, 16810–16816). That is why, we resorted to scanning tunneling microscopy (STM) for atomic resolution of the structure, which was much more difficult and time consuming.

Figure R2 | Scanning electron microscopy (SEM) images of the F-COF. a-d, SEM images at different magnification.

Figure R3 | Transmission electron microscopy images of the F-COF. a-b, Low resolution and c-d, High magnification TEM images.

Comment 1.9. The text in the experimental section of SI “Important Note regarding STM” should be refined to make it more clear and improve English language usage. How are the samples prepared for STM? Is there sonication and dropcasting involved? What are the key instrument settings for obtaining the images?

Response 1.9. Thanks to the reviewer for highlighting the poor writing issue. Now, we have modified the text to make it more understandable for the readers. The sample preparation procedure is already given in the ‘Methods section’ of the manuscript. In short, the powder sample was directly loaded on the sample stage of the STM and sublimized *in situ* inside the chamber and characterized.

Comment 1.10. What is the red trace in Figure S1? Please label.

Response 1.10. Sorry for the confusion. The red trace was related to heat flow changes with respect to temperature. To avoid confusion, we have removed the red trace in the TGA thermogram.

Reviewer #2(Remarks to the Author):

This work describes the formation of pyrazine consists of a reversible amine/ketone addition and an irreversible ring closure. The value of the work is that the product is much more conjugated than normal Schiff-base COF, so this represents a significant advancement.

Comment 2.0. One point that needs attention is that the crystallinity is more like a polymer from powder XRD. Can the authors comment on this? compared to the usual COF made the XRD peaks of this is broad, should it be closer to a polymer when crystallized in bulk form.

However on-surface synthesis shows that 2-D COF can be formed locally. The STM result is reasonably good among all published 2D polymer grown in STM, in terms of uniformity and size. Such dehydration reaction is usually not suitable for on-surface growth, so this result is excellent and the work can be published. At least, it provides a design guiding principle for future researchers.

Response 2.0. We greatly appreciate the reviewer #2 for his/her thoughtful evaluation and recognition of the importance of this work. As a matter of fact, we had the same opinion with the reviewer, because of the presence of broad XRD peaks. However, after resolving structure using STM, we concluded that we were able to produce a right material and the XRD peak broadening should be associated with not poor ordering but poor packing. Sorry for the confusion, we did not prepare the structure by reacting the monomers on the surface. Instead, we prepared the structure in solution and later resolved the structure on the Cu(111) substrate by subliming the prepared material under ultrahigh vacuum chamber.

Reviewer #3 (Remarks to the Author):

In this manuscript, the authors synthesize a covalent organic framework featuring fused aromatic rings via irreversible condensation. This pyrazine-based COF has been characterized with different techniques (PXRD, TGA, XPS, FT-IR, STM etc.) and exhibits a high physicochemical stability owing to its fused aromatic ring system.

After reading through the whole manuscript carefully, I think it is not suitable to publish in Communications Chemistry as it does not meet the novelty criteria and the studies are not sufficient.

Comment 3.1. Compared with other pyrazine-based COFs (J. Am. Chem. Soc. 2019, 141,

16810., ref. 42), the quality of PXRD pattern is poor, and the values of Rwp and Rp are not reasonable.

Response 3.1. We thank the reviewer #3 for valuable suggestions to improve the quality of the manuscript. Somehow, we agree with the reviewer's concerns, but the values of Rwp and Rp are not so critical in the XRD simulation, in the Pawley refinement, if the crystal size is varied before simulation, the Rwp and Rp values also change (increase or decrease). Based on the reviewer's concern, we rechecked the Pawley refinement results. Thanks to the reviewer. Now, the results are more refined with better fit and lower Rwp (3.48%) and Rp (5.44%) values. Accordingly, we have changed the Pawley refined curve in the graph (**Figure 2a**). The most important thing in the experimental data is (100) peak, which indicates the crystallinity of the material, but the peaks are broad because of various reasons, including small crystal size, polycrystallinity, poor and mismatched staking of the layers and so on (**Also see Response 1.4**). Nevertheless, STM images clearly reveal the structure of this material.

Comment 3.2. Compared with the paper that the authors published in Nat. Commun. previously, this manuscript does not have too much advantage and novelty.

Response 3.2. We partially agree with the reviewer. The pyrazine chemistry as polymer forming reaction is the same with Nature Communication's paper. However, the structure in this work is totally different. F-COF contains free ($-NH_2$ and $-OH$) functional groups, which can be post modification potentials to tune many other properties, including electronic properties and solubility in organic solvents (**Figure R1**).

Comment 3.3. COF is crystalline polymer and not easy to sublime. For the STM experiment, the authors have claimed that "They rigorously evaporate and sublime.". Why? Are they oligomers? Or they are the mixture of COFs and oligomers?

Response 3.3. We highly appreciate the reviewer #3 for thoughtful comments. We are sorry for the confusion. Let us explain in detail for the reviewer's better understanding. The single layer was evaporated thermally at 600 K on cleaned Cu(111) substrate under ultra-high vacuum (UHV) condition. As the reviewer may well understand that macromolecular materials, which are synthesized by solution polymerization, have a wide range of molecular weight distribution from small to very large sizes. It is almost impossible to synthesize macromolecular materials with the same molecular size and uniform molecular weight distribution. Thus, under ultrahigh vacuum condition, it is possible to sublime relatively smaller sized flakes onto the clean Cu(111) substrate. We also observed different flake sizes with STM but we did not see any oligomeric uncyclized fragments.

Comment 3.4. The Solid state NMR of the COF should be provided to analyze the structure of F-COF.

Response 3.4. On the reviewer's suggestion, we checked the solid-state NMR of the F-COF. The solid state ^{13}C -NMR reveals peaks related to edge diketones at 176.40 ppm and aromatic carbons centred at 138.50 ppm. These broad peaks are matching well with the proposed structure (**Figure R4**).

Figure R4 | Solid state ^{13}C -NMR spectrum indicating peaks related to the structure.

Comment 3.5. There are some mistakes in the manuscript and SI, such as Supplementary Fig. 1, weight loss is wrong.

Response 3.5. We made necessary changes to the supplementary data according to reviewer's suggestion.

REVIEWERS' COMMENTS:

Reviewer #1 (Remarks to the Author):

I appreciate the author's attempt to address the criticisms of this reviewer. However, I have found that on several occasions while the authors went to great lengths to explain various issues to this reviewer, the manuscript was not really improved, or no modification to the manuscript were made in response to the comments. Provided that the manuscript is modified in response to reviewers' comments to a greater extent, I believe this work can be published after minor revisions.

For instance, for comment 1.4, rather than writing to the reviewer the authors should re-think how they might clarify a similar question to the actual reader of the manuscript after it is published.

For comment 1.5, the rationale for any solvent choices should be captured in writing in the supporting information.

For comment 1.6, the authors should consider adding Figure R1 to the supplementary information, where it can serve to clarify structural issues to the reader.

For comment 1.8, the authors should show all the images that they presented in their rebuttal letter to the readers, not just the select few.

Reviewer #2 (Remarks to the Author):

The authors have addressed all my previous question and i am happy to recommend the work for acceptance.

Reviewer #3 (Remarks to the Author):

It can be accepted.

Reviewers' comments:

Reviewer #1 (Remarks to the Author):

I appreciate the author's attempt to address the criticisms of this reviewer. However, I have found that on several occasions while the authors went to great lengths to explain various issues to this reviewer, the manuscript was not really improved, or no modification to the manuscript were made in response to the comments. Provided that the manuscript is modified in response to reviewers' comments to a greater extent, I believe this work can be published after minor revisions.

Comment 1.1: For instance, for comment 1.4, rather than writing to the reviewer the authors should re-think how they might clarify a similar question to the actual reader of the manuscript after it is published.

Response 1.1: We greatly appreciate the reviewer #1 for the constructive comments to improve the quality of this work to next level. As the reviewer may already know these are some common facts regarding the XRD application to organic frameworks. Keeping in view of the reviewer's suggestion, we have added additional clarification in the manuscript (**Supplementary Note 1**).

Comment 1.2: For comment 1.5, the rationale for any solvent choices should be captured in writing in the supporting information.

Response 1.2: The choice of solvent with enhanced description has been incorporated in the revised manuscript.

Comment 1.3: For comment 1.6, the authors should consider adding Figure R1 to the supplementary information, where it can serve to clarify structural issues to the reader.

Response 1.3: Based on the reviewer's suggestion, we have added the Figure R1 in the revised manuscript as a **Supplementary Fig. 10**.

Comment 1.4: For comment 1.8, the authors should show all the images that they presented in their rebuttal letter to the readers, not just the select few.

Response 1.4: We have added all the images (SEM and TEM), which were shown in the point-to-point response letter in the revised manuscript based on the reviewer's suggestion (**Supplementary Fig. 5 & 6**).

Reviewer #2 (Remarks to the Author):

The authors have addressed all my previous question and i am happy to recommend the work for acceptance.

Response: We appreciate the reviewer #2 for the invaluable comments to improve the quality of the manuscript.

Reviewer #3 (Remarks to the Author):

It can be accepted.

Response: We appreciate the reviewer #3 for the precious comments and suggestions to enhance the quality of the manuscript.